# The Evolution of Large Organism Size: Disparate Physiologies Share a Foundation at the Smallest Physical Scales

**DOI:** 10.3390/life15121914

**Published:** 2025-12-14

**Authors:** Simon Pierce

**Affiliations:** Department of Agricultural and Environmental Sciences (DiSAA), University of Milan, Via Celoria 2, 20133 Milano, Italy; simon.pierce@unimi.it; Tel.: +39-02-503-16785

**Keywords:** systems biology, transport networks, diffusion, bulk flow, multicellularity, cytoplasmic streaming, evolutionary transition

## Abstract

Life is defined by self-governing networks of molecules that change conformation cyclically, converting thermodynamic motion into directional work and structure. A spectrum of scale, from nanoscopic to macroscopic, involves a shift from intracellular thermodynamically driven processes (thermal agitation ultimately rooted in quantum phenomena) to intercellular bulk flows described by classical physics; from short-distance transport involving diffusion and cytoskeletal transport to long-distance pressure fluxes in hydraulic networks. A review of internal transport systems in macroscopic eukaryotes suggests that a key evolutionary step favoring large size and multicellularity involved exploiting molecular-scale stochasticity to generate organized bulk flows (e.g., motor proteins collectively generating mechanical pressures in metazoan tissues such as cardiac muscle; within tracheophytes, active and passive phloem loading/unloading inducing pressure gradients, and active regulation enabling passive xylem function and hydraulic reliability; sieve-like conduction in heterokonts; and peristaltic shuttle streaming in myxogastrian plasmodia). Macroscopic physiologies are underpinned by Brownian molecular thermodynamics and thus quantum mechanics; the apparently disparate physiologies of large organisms share a fundamental operating principle at small scales. However, the specific translocation mechanisms that extend this functioning to larger scales are embroiled in bauplans, representing phylogenetic constraints to body size.

## 1. Introduction

Living systems are self-governing networks of components that work to create structure; macromolecules changing conformation dependably, driven by thermodynamic gradients to induce directional processes and work that consolidates matter [1,2,3,4]. A fundamental requirement for the operation of these networks is motion. This includes the characteristic uniplanar motions (ratcheting or spinning [4,5,6,7]) of the single macromolecules and multi-subunit complexes, and also the fluidity of membranes and the internal transport of substrates and products to and from different regions within the organism. Constraints to the transport of material within organisms determine the capacity to scale in size and complexity [8,9] and are a key issue that different taxonomic groups have evidently overcome in contrasting ways during the evolutionary history of life on Earth.

Indeed, the evolution of large size, in both unicellular and multicellular organisms, involved surmounting the excessive time involved for transport over relatively large internal distances when relying on processes driven solely by thermodynamics (i.e., physical work/energy relationships). Specifically, transport times represent a critical limitation to the supply of substrates to metabolism and the elimination of waste products and, thus, physiological integration [10]. In terms of the physical forces that drive transport processes, the common underlying problem faced by all groups of organisms is the transition from extremely small-scale nanoscopic networks of molecules (in which thermodynamics and causal quantum mechanical processes determine motion) to the macroscopic, in which movement of bulk matter obeys the rules of classical physics and is essentially determined by pressure gradients.

## 2. The Quantum to Classical Boundary and Its Relevance to Biology

Strictly—as understood in physics—the quantum to classical boundary represents the threshold between observations that are described by quantum rules and those described by classical laws. This is of relevance to biology because it explains how the matter from which organisms are constructed behaves at a fundamental level. Here, a ‘quantum-to-classical transition’ in physiology specifically denotes the threshold at which molecular processes (dominated by thermodynamic, stochastic processes emerging from quantum mechanics) give rise to bulk properties describable by classical physics (fluid dynamics). To understand this transition, it is first important to consider how matter is interpreted in the context of quantum mechanics. Quantum mechanics describes particle superpositions, interference, and entanglement [11,12,13,14], while classical mechanics depicts definite states and trajectories, and evidently the classical world emerges from the quantum realm. For example, an electron (a localized excitation in the quantum field permeating the universe) is described by a wavefunction that gives the probability of finding it in a particular state or position [14]. But until it is observed (i.e., interacts with something) the electron effectively exists in superpositions of states—in multiple places or momenta at once—of which we cannot be certain. The same ‘quantum uncertainty’ applies to the subatomic particles making up atomic nuclei, and even to the state of the entire atom, meaning that in practice when atoms are at rest they oscillate and vibrate in place (even at 0 °K; hence the term ‘zero-point energy’ [15]). Atoms also possess kinetic energy and, especially when in solid lattices, transfer this to one another by virtual particles called phonons (i.e., they jostle one another, with the phonon representing the interaction). In real terms, atoms and molecules spontaneously jiggle (an informal but serious term often used by Richard Feynman to explain a phenomenon that is difficult to encapsulate in English [16]) with a combination of thermal (kinetic) and zero-point energy. This is the quantum mechanical basis of emergent random particle motion (thermal agitation or Brownian motion [17]) and is ultimately the physical driving force of classical physical processes such as diffusion along concentration gradients [18,19].

Another link between quantum and classical worlds, and how this relates to the physical scale and size of organisms, is decoherence, or the cancellation of quantum effects when particles interact with (are ‘observed by’) their environment [20,21,22]. Decoherence is faster for systems of larger mass because larger numbers of atoms participate in more interactions with atoms and photons in the environment. Thus, as mass increases, decoherence is more likely (coherence time and quantum effects decrease), and a system moves along a continuum from quantum to classical. In biological systems, single macromolecules in the ‘warm–wet’ environments inside organisms are at the upper mass limit for decoherence. Nonetheless, and despite the energetic environmental context, the atoms comprising them can exhibit coherence and quantum mechanical effects, e.g., [23,24,25,26], so much so that ‘quantum biology’ is now a recognized subdiscipline [27,28]. Essentially, their motion is an emergent effect of their own inherent thermal agitation, but particularly of bombardment by smaller thermally agitated molecules such as those of water. In this thermal bath, macromolecules and molecular subunit complexes such as ribosomes, RNA polymerase, motor proteins, and other enzymes continuously explore a broad conformational energy landscape (i.e., change shape semi-randomly depending on interactions with the environment), with fluctuations and transitions that are essential for their function. Rather than being hindered by thermal noise they harness it, converting random motion into directional motion and reliable activity [2,3,4,5,6]. Thermal agitation of molecules and diffusion gradients (especially when energetic barriers are overcome by energy inputs such as excitation donated by photons during photosynthesis) are the fundamental physical processes driving metabolism and ultimately emerge from quantum mechanical effects (Figure 1).

Note that the ‘energy carrier’ nucleotide adenosine triphosphate (ATP) is created by transmembrane electrochemical gradients driven mainly by diffusion; it carries energy derived from thermodynamic gradients, and so from thermal agitation, Brownian motion and ultimately from the quantum uncertainty that underpins atomic and molecular motion (Figure 2). Note also that other nucleotides, such as guanosine triphosphate (GTP), are synthesized by enzymes driven by ATP hydrolysis, and so essentially carry energy previously carried by ATP and thus again ultimately from thermodynamic disequilibria.

## 3. Biological Phenomena Driven by Stochastic Thermodynamic Effects

Enzymatic catalysis exemplifies the role of thermal agitation in function. Enzymes undergo thermally driven conformational changes that align catalytic substrates into productive geometries, transiently stabilizing transition states and lowering activation barriers [29,30]. ATP is required to overcome energy barriers, inducing suitable conformation changes that essentially activate the enzyme, but enzyme motion during function is also driven directly by thermal agitation and bombardment [4,5,6]. Dependence on thermal agitation appears to be true of all catalytic globular protein complexes [31,32,33]. For example, RNA polymerase relies on stochastic conformational changes for its operational movement along the DNA template during transcription; local thermal fluctuations enable forward and backward motion of the polymerase, with nucleotide incorporation (of ATP, but also GTP, UTP or CTP depending on the DNA template) biasing the process toward net elongation (i.e., RNA polymerization) [34]. Note that in this case nucleotides bias the direction of the process, but ATP is a component incorporated into RNA rather than simply being an energy carrier—the main source of energy is thermal agitation and bombardment (thermodynamics).

Similarly, the ribosome, a large ribonucleoprotein complex, exhibits extensive Brownian motions during translation, including inter-subunit rotation [35,36,37,38,39]. These thermally accessible conformational changes, coordinated by GTP hydrolysis, ensure the fidelity and directionality of protein synthesis.

Eukaryotic motor proteins such as myosin, kinesin, and dynein exemplify the conversion of energy into mechanical work under conditions dominated by thermal bombardment. Their movement along actin filaments or microtubules is inherently stochastic, consisting of cycles of binding, conformational change, and diffusion. ATP hydrolysis biases these thermally driven fluctuations, contributing to net directed motion. The coupling between random molecular motion and energy rectification, often modeled as a Brownian ratchet, underscores the fundamental principle of molecular biophysics that thermal noise is not suppressed but harnessed to perform work with high efficiency [3,5,6,34,35,36,37,38,39]. Note that within myocytes (contractile muscle cells), Brownian motion and ATP hydrolysis induce the single myosin molecules to pull against actin, evident collectively at the macroscopic scale literally as the physical force with which the muscle contracts [40].

Another example of a thermally coupled molecular machine is the rotary enzyme ATP synthase. This complex, found in mitochondria, chloroplasts, and bacterial membranes, operates as a reversible rotary motor (Figure 1). Diffusion of protons through the membrane-embedded *F*_0_ motor generates torque, driving rotation of the central stalk [41]. This pushes against subunits of the catalytic *F*_1_ region, inducing “hinged” [42] conformational changes central to synthesizing ATP from ADP and inorganic phosphate. Conversely, hydrolysis of ATP can drive reverse rotation, with the enzyme functioning as an ATP-driven proton pump. Thermal fluctuations facilitate subunit rotation and conformational transitions across energy barriers. In essence, the proton-motive force does not impose deterministic motion but rather biases the probability distribution of rotational states, leading to net directional rotation along one plane and broadly linear conformation cycles in catalytic *F*_1_ subunits. A similar conformational and energetic mechanism is apparent for ATPase, which typically uses ATP to drive rotational motion, setting up electrochemical gradients [43,44] (N.B.: structurally, ATP synthase is considered a subclass of ATPases, with the suffixes referring to different primary functions; essentially tending to produce or to use ATP). The operation of ATP synthase and ATPase enzymes exemplify the principle that biological nanomachines exploit thermal agitation as an integral component of their mechanochemical cycles.

## 4. Diffusion Limitation to Organism Size

Diffusion, governed by Brownian motion and thermodynamics, dominates at small scales. But as volume increases, the time for diffusion of a molecule across a body grows approximately with the square of the distance while the capacity for uptake via surface area grows linearly, thus imposing a surface-area-to-volume constraint on size. For example, in plants manipulation of Fick’s second law [45] demonstrates that beyond a certain size, passive diffusion is insufficient and bulk-flow systems become necessary. Indeed, without the evolution of systems for the translocation of volumes of fluids (bulk flow), plants would have remained only a few centimeters tall [46,47,48]. Conversely, advection and bulk flow facilitate large size by distributing nutrients, signaling molecules and wastes more rapidly than diffusion alone.

Before true bulk-flow vascular systems can evolve, organisms exploit intermediate mechanisms. For instance, cytoplasmic streaming is a large-scale movement of cytoplasm in eukaryotic cells, induced by molecular motor systems working on the cytoskeleton, to overcome diffusion limitations in cells of unusually large size. Cytoplasmic streaming and bulk intercellular flow become necessary when cell diameters exceed ~100 µm; a value determined by diffusion that holds true for all major eukaryotic clades [49]. At even smaller scales, intracellular transport on the cytoskeleton blends diffusion and directed motion: cytoskeletal network morphology strongly regulates intracellular transport dynamics and times [50,51]. Thus, within cells we see a gradient from diffusion, motor-driven transport through to streaming/advection in the liquid cytoplasm, which is essentially a small-scale bulk flow driven not by pressure gradients but propelled by the collective motions of hundreds of thousands (to millions, depending on cell type) of working molecules.

Crucially, the evolution of multicellularity extends these cellular transport modes beyond the single-cell scale. A key stage in the evolution of multicellularity was the evolution of colonial life, and this involves intermediate transport mechanisms external to single cells. In green algae of the order Volvocales (e.g., *Volvox carteri* F.Stein; Figure 2), the coordinated beating of flagella at the colony surface generates fluid flow that enhances nutrient uptake: in this system advection (transport via bulk motion of fluid) produces a nutrient-exchange rate that is quadratic in radius, overcoming the diffusion limitation locally. This illustrates an early form of bulk flow at the colony scale (i.e., ‘quasi’ multicellular). The transition to full-blown hydraulic (pressurized) systems in various clades also involves the intermediate evolutionary step of pseudo-circulatory mechanisms inducing local bulk flow in internal volumes of liquids.

## 5. Internal Long-Distance Transport Systems in Fungi

Of the eukaryotes, the first taxa known to use long distance, cell-to-cell transport were the fungi, evolving these phenotypes (and by inference these physiological systems) potentially as early as ~1.4 Ga (billion years ago) [52]. In filamentous fungi, hyphae are composed of cells connected end-to-end and compartmentalized by perforated septa, with the cytoplasm being largely continuous across the colony. This configuration produces a multinucleate, ‘syncytial’ cytoplasm (a continuous intercellular protoplast) through which water, ions, metabolites, and organelles can move freely [53,54]. In many fungi, anastomoses (hyphal fusions) further increase cytoplasmic connectivity, enabling extensive internal transport and resource sharing across the network. Passive transport processes within the fungal syncytium are primarily based on diffusion and pressure-driven bulk flow. Small solutes such as sugars, amino acids, and ions can diffuse passively through the cytoplasm and across septal pores following concentration gradients [55]). In addition, differences in hydrostatic pressure (turgor) between hyphal compartments may generate cytoplasmic streaming and bulk flow, which facilitate long-distance transport of solutes and water [56]. This is somewhat analogous to the pressure-flow mechanism of phloem transport in plants (see Section 8), although in fungi the driving forces are localized and transient rather than maintained by specialized source-sink relations.

Active transport processes provide regulation and directionality within the syncytium. Membrane transport proteins, including proton pumps (H^+^-ATPases), maintain electrochemical gradients that energize the secondary active uptake or efflux of nutrients such as sugars, amino acids, phosphate, and metal ions [57,58]. This energy-dependent ion transport contributes to the maintenance of internal pH and turgor pressure homeostasis, both of which are essential for hyphal growth and intracellular circulation [53]. Furthermore, vesicle trafficking and organelle movement within hyphae rely on cytoskeletal transport using motor proteins such as myosins, kinesins and dyneins, which actively transport cargo along actin filaments and microtubules [59,60]. These mechanisms facilitate the targeted delivery of enzymes, lipids and wall-building materials to growing hyphal tips, as well as the redistribution of metabolites from older to actively extending regions of the colony. The integration of active and passive transport ensures that the fungal syncytium functions as a coherent and adaptive physiological system. Energy-demanding ion transport and motor-driven movement establish and maintain turgor and concentration gradients; active generation of conditions that induce passive flows.

## 6. Internal Transport in Plasmodial Myxogastria (Slime Molds)

The crown-group of the Amoebozoan class Myxogastria (plasmodial slime molds) diverged at an estimated 823–661 Ma [61]. The vegetative or plasmodial stage of the slime mold *Physarum polycephalum* Schwein exhibits an exceptional form of internal transport that enables coordination and resource distribution across its macroscopic body, which may span several centimeters to meters in size (networks of ≥10 cm^2^ have been investigated in the laboratory). In the plasmodial stage, *Physarum* exists as a single, multinucleate cell organized into a dynamic network of tubular veins through which cytoplasm is rhythmically shuttled in alternating directions in a process known as ‘shuttle streaming’. Unlike cytoplasmic streaming in the organisms considered above, movement is driven by periodic contractions of an actomyosin cortex in the vein walls (in other words, pressure waves that propagate through the system, induced and coordinated by a tubular arrangement derived from the cytoskeleton). This sets up peristaltic pressure waves, propelling cytoplasm back and forth throughout the network as a bulk flow at speeds of up to 0.048 mm s^−1^ [62,63]; slow compared to transport systems in other eukaryotes, but nonetheless an order of magnitude faster than diffusion for large molecules such as proteins.

The mechanism underlying this transport is biophysical and biochemical: contractions of the actin–myosin system are regulated by calcium concentrations, forming a Ca^2+^-dependent contractile oscillator that sets the frequency and amplitude of streaming [64]. Oscillations propagate as phase waves along the network to create peristaltic motion ensuring efficient advection of solutes and organelles over macroscopic distances [65]. Experimental and theoretical studies have demonstrated that the plasmodium regulates its contraction wavelength and phase relationships depending on its current size to optimize fluxes [62]. The resulting peristaltic flows achieve long-range transport far more effectively than diffusion alone, mixing and distributing metabolites across the entire organism [65,66].

Internal flows also serve a signaling function. Changes in environmental conditions such as nutrient availability or light stimuli induce localized alterations in contraction amplitude and frequency, which can propagate as waves of enhanced or suppressed contraction through the network, effectively transmitting information about the external environment [62]. Thus, the same physical processes that drive nutrient transport also underlie a primitive form of distributed physiological coordination.

The macroscopic size of plasmodial slime molds is fundamentally constrained by the mechanics of their internal transport system. Experimental and theoretical studies demonstrate that the wavelength and frequency of contractions during shuttle streaming must scale with organism size to maintain effective advective transport; mismatches between contraction patterns and body geometry reduce transport efficiency and can limit the functional size of the plasmodium [62,64]. Furthermore, physical constraints such as vein diameters and viscous dissipation set limits on transport rates, and thus on how large a single, integrated plasmodium can grow without altering its network architecture [64,65]. Computational and analytical models further support the conclusion that organismal size is limited by transport optimization, as network geometry and flow dynamics determine the maximum scale at which shuttle streaming can maintain physiological integration [62,64]. This system represents the upper size limit and greatest sophistication of long-distance transport within single-celled organisms, beyond which large size requires dedicated cell types and multicellular tissues.

Thus, like the fungi and other eukaryotes, the translocation system has a basis in thermodynamics and molecular actions that then create classical bulk flows at the macroscopic scale. This overcomes diffusion limitations and facilitates large size, but the transport distances that can be sustained by shuttle streaming are limited relative to the distances evident in larger multicellular eukaryotes with dedicated transport tissues. In other words, slime molds demonstrate that translocation and size are phylogenetically constrained by bauplan.

## 7. Origin of Circulatory/Vascular Systems in Metazoa

The earliest multicellular animals (e.g., sponges, simple cnidarians), diverging 613.2 to 593.4 Ma in the Ediacaran period [67], lacked a dedicated vascular system; their tissues remained in direct contact with seawater or water within cavities, allowing diffusion-driven exchange [68,69,70]. With the evolution of triploblasty (i.e., three germ layers giving rise to associated tissue systems) and larger body size, animals required more effective internal transport. Comparative analyses suggest that a rudimentary blood vascular system first appeared in an ancestral bilaterian (581.8–569 Ma [67]), as an adaptation that overcomes the diffusion limit over larger transport distances [70,71]. This ancestral system likely consisted of coelomic cavities or channels lined by mesothelium rather than a true endothelium, combining features of lacunar and vessel-lined systems [68]. Multiple independent acquisitions of closed or semi-closed vascular systems are inferred, consistent with parallel evolution of circulatory architectures in diverse phyla [68]. The evolution of an endothelial lining occurred later in the vertebrate lineage (~420 Ma [66]), a step facilitating higher perfusion pressure (i.e., the driving force for blood flow), greater barrier control, and functional specialization [69,70,72]. Thus, vascular evolution may be viewed as a sequence from diffusion-based transport, through lacunar/coelomic transport, primitive vascular networks, to closed endothelial-lined systems.

Specifically, as diffusion (particularly of oxygen) becomes limiting at distances greater than around 100 µm [49], cells such as *Amoeba* and *Paramecium* (cell diameter ~10–100 µm) are small enough to rely solely on diffusion. However, at the scale of nematodes (such as *Caenorhabditis elegans* Maupas 1900; ~0.5–1 mm), diffusion is insufficient and a circulatory mechanism for pseudocoelomic fluid is required, based on muscular movement of the trunk which induces internal fluid movement. True, closed circulatory systems later evolved in annelids, mollusks, arthropods and vertebrates (Table 1).

Invertebrates exhibit a wide variety of circulatory systems, broadly categorized as open (hemocoelic) or closed (vessel-based) systems, though this distinction is often gradational [72]. In many arthropods (insects, crustaceans) and non-cephalopod mollusks, a contractile dorsal vessel pumps hemolymph into a hemocoel bathing the organs before returning to the heart via ostial openings. This arrangement is sufficient for small organisms or moderate activity levels [73]. In contrast, annelids possess a closed vascular system with dorsal and ventral longitudinal vessels and segmental interconnections [70]. In some species, such as *Nereis japonica* Izuka 1908, the vessel walls lack a discrete endothelial layer and resemble interstitial spaces [73]. Pan-arthropods (Panarthropoda) such as onychophorans (“velvet worms”) show open vascular systems with peristaltic hearts and segmental ostia; this is probably the ancestral condition for arthropods [74].

Across invertebrates, the shift from diffusion to vessel-based transport correlates with increases in body size, metabolic rate, activity level, and body plan complexity [70]. Furthermore, vascular evolution is intertwined with respiratory pigment evolution. The marine annelid *Platynereis dumerilii* Audouin & Milne-Edwards 1834 exhibits a closed vascular system with extracellular hemoglobins; genomic analyses reveal that the ancestral bilaterian possessed at least five globin gene clades, indicating multiple independent recruitments of respiratory pigments alongside vascular innovations [75]. Thus, invertebrate vascular systems illustrate diverse evolutionary adaptations, from open hemocoelic systems to rudimentary vessel networks, contingent on the physiological demands of ecological circumstances.

Within deuterostomes (bilateral animals in which embryo development is characterized by the anus preceding gut and mouth development), the evolution of closed blood-vascular systems culminated in vertebrates with endothelial-lined vessels, muscularized arterial walls, capillary beds for exchange, and venous return systems [69]. In the vertebrate stem, the appearance of endothelial cells permitted higher intravascular pressures and controlled permeability [70]. Subsequent vertebrate diversification involved progressive cardiac specialization: fish generally exhibit a single-circuit, two-chambered heart; amphibians and most reptiles exhibit a partially divided three-chambered heart; and birds and mammals evolved a fully separated four-chambered heart supporting dual high- and low-pressure circuits [2,76]. Additional innovations include capillary exchange networks and arteriolar smooth muscle regulation and lymphatic vasculature.

Some detail is known regarding the specific steps involved in the evolution of metazoan cardiovascular systems. A recurring theme is the modular reuse and quantitative tuning of conserved genes and pathways. Rather than representing the expression of novel genes, morphological novelties such as additional heart chambers, specialized conduction tissue and high-pressure arterial systems generally reflect shifts in spatial/temporal expression, gene dosage, cis-regulatory reorganization (i.e., changes in interactions between non-coding DNA and nuclear proteins regulating gene expression) and paralogous gene use (i.e., genes that at some point were duplicated and subsequently became entrained in novel functions). Signaling pathways (specifically BMP, FGF, NOTCH, VEGF) and transcriptional modules (master regulators of cardiogenesis, such as NKX2-5, TBX, GATA, MEF2, ETS, HAND) together form the developmental ‘scaffolding’ modified by natural selection to result in diverse heart morphologies across vertebrates [77]. For instance, the heart evolved from a muscular tube and acquired modules such as atria, ventricles, septa and valves. These evolutionary steps are evident during the embryonic development of vertebrate hearts (e.g., the progressive appearance of the cardiac crescent, linear heart tube, looping heart, chambers and septa during the first 12 d of mouse heart development) and the transcription factors involved in these steps are either known or suspected [77]. Note that the same transcription factors may be produced in diverse vertebrate clades but control very different aspects of heart development, acting to regulate different target genes or interacting in combination to produce distinct effects [77]; evidently, there is no simple or single developmental journey (nor presumably a simple series of evolutionary events) for animal circulatory systems.

However, there is a simple shared operating principle underpinning function. The pressure within animal circulatory systems is generated by muscular activity, either within the cells (of cardiac tissue) within a heart or the muscles responsible for moving the main trunk of the body. Muscles operate due to cytoskeletal activity in which motor proteins are driven by thermodynamic gradients (both direct thermal agitation and from ATP or GTP, which essentially carry energy derived from thermodynamic gradients—GTP is synthesized using ATP, and so indirectly carries the energy from the original transmembrane electrochemical gradient that produced ATP). These are ultimately driven by Brownian processes and, thus, like the other groups of eukaryotes discussed above, also conform to the concept of thermodynamics and molecular action driving bulk flow transport systems, to overcome diffusion limitations and facilitate large organism size.

## 8. Viridiplantae: Active Regulation of Passive Fluxes

Photosynthetic organisms using chlorophylls a and b (Viridiplantae) include the green algae (Chlorophyta) and the land plants (Embryophyta; with differentiated embryos). Giant coenocytic (i.e., single-celled multinucleate) green algae, such as *Caulerpa* species, can reach sizes of up to 2.8 m in length [78] but the entire thallus, including stolons, upright fronds and rhizoids, is a continuous cytoplasmic mass enclosed by one plasma membrane and a cell wall, but lacking any internal dividing cell walls. It uses cytoskeletal transport to induce cytoplasmic streaming, essentially driven by myosin motor proteins acting on actin filaments. In large specimens, streaming ensures physiological integration between photosynthetic and absorptive regions of the alga. In effect, cytoplasmic streaming in *Caulerpa* serves a role analogous to the vascular transport of plants, maintaining metabolic coordination and enabling rapid redistribution of resources across its coenocytic body, but it occurs within a single general-purpose, multi-role cell, lacking a dedicated translocation cell-type or tissue. As with animals, the macroscopic movement is ultimately generated by molecular motion and thermodynamics occurring within cells.

The earliest embryophytes colonized the terrestrial surface probably around 515.5 to 469.0 Ma (middle Cambrian–Early Ordovician; [79]). They lacked vascular tissues and were physically limited to the soil/atmosphere interface by a reliance on diffusion. The evolution of simple vascular cell elements in some polysporangiophytes (branching plants such as *Cooksonia pertoni* Lang) were associated with spore-bearing bodies held aloft to facilitate spore dispersal [80], leading to the development of photosynthetic stems and true roots, and granting access to water and nutrients at depth within the soil. These plants were characterized by specialized translocation tissues, the xylem and phloem, which are retained in extant vascular plants (Tracheophyta or tracheophytes), which diverged ≈ 450.8–431.2 Ma [79].

During xylem function, water movement from the soil, through roots and stems to the atmosphere, occurs essentially along gradients of water concentration moderated by surface contact (matric) interactions; expressed as the tendency of water to move (or ‘water potential’; Ψ). Movement is driven largely passively by classical physical forces. However, water potential is subject to biological regulation at key cellular and tissue levels which actively improves function and performance, and is crucial to ‘kick-starting’ the process following stress events or damage. Water diffuses from the soil, passes through the root tissues into the xylem, ascends the stem, enters leaf tissues and finally evaporates into the atmosphere (transpiration). The overall flux proceeds from regions of higher water potential to regions of lower potential, and accordingly is most often described as a passive process. Indeed, the ascent of sap in the xylem is due to the cohesion–tension mechanism, which relies on transpiration to generate negative pressure (tension) on a cohesive column of water in the xylem vessels, pulling water upwards against gravity. Water enters root hair cells and the root cortex by osmosis and matric potential gradients, with negligible requirement for metabolic energy. Once water enters the xylem, its movement upward is largely passive.

However, although the bulk of the water flux is passive, plants do exert metabolic control at two principal regulatory sites: the root and the leaf. Root cells actively transport ions (e.g., K^+^, NO_3_^−^) into the vascular tissues, thereby lowering the local water potential and promoting osmotic water uptake. While the water movement itself remains passive, the gradient that induces it is metabolically generated (especially so after periods of drought, when cavitated xylem vessels must be re-filled to re-establish the soil–plant–atmosphere continuum and restart the cohesion–tension mechanism). Moreover, the presence of apoplastic barriers (i.e., obstacles to water movement within the structure of the cell wall) such as the endodermis and corresponding hydrophobic Casparian strip forces water and solutes to traverse membranes, which are key points of control enabling selective uptake of beneficial nutrient ions and exclusion of deleterious ions such as salt or toxic metals. Aquaporins (water channels) function passively via osmosis, but are regulated actively by phosphorylation, pH, and [Ca^2+^], which are under metabolic control. This in turn regulates root hydraulic conductivity in response to environmental and internal cues. Additionally, at the leaf surface the rate of water loss (and hence the driving force for ascent) is regulated by the active opening and closing of stomata. Guard cells use ATP to pump ions (via H^+^-ATPases, K^+^ channels) and adjust their turgor to determine stomatal aperture: an active process modulating the passive flux of water vapor.

Thus, while the driving force sensu stricto for xylem operation is passive, it is actively regulated, and this has a basis in thermodynamics and the ‘machinery’ of the cell. Plants expend metabolic energy to generate osmotic gradients in the root, to modulate membrane and channel properties, and to regulate stomatal aperture. Water fluxes are embroiled in a regulated framework ensuring homeostasis, preventing uncontrolled influx or efflux and allowing responsiveness to environmental changes (e.g., soil moisture, salinity, and vapor-pressure deficits). This duality of passive transport with active regulatory input is critical: purely passive systems would lack selectivity or dynamic adjustment, while purely active pumping of water would be energetically unfeasible given the volumes and heights evident in plants.

Additionally, note that plant height does not have a simple relationship with xylem conduction, and complicating factors can contribute to a large size. As an extreme example, the tallest tree, a *Sequoia sempervirens* (D.Don) Endl. individual known as ‘Hyperion’ growing in Redwood National Park, California, USA, is 116.07 m tall. However, the relationship between height and internal water translocation is complicated by the fact that the upper leaves of this species (particularly on the main axial branches) can absorb water from mist and rain (indeed, they grow in cool humid coastal climates where fog and low cloud readily form). They also have a relatively large volume of parenchyma tissue dedicated to water storage. This decreases the reliance of photosynthesis and growth (and thus of size) on transport from the roots [81,82]. The canopy of such trees can also accumulate detritus, encouraging soil formation, which is potentially a further source of nutrients independent of the xylem.

Crucially, within root and stomatal cells, the membrane-associated molecules involved in active regulation of xylem sap translocation are subject to Brownian motion and thermal agitation, and indeed rely on these for their operation. This classical, large-scale and (for the most part) passive translocation system can only function effectively due to molecular systems operating on underlying thermodynamic effects.

In plant phloem, the translocation of organic solutes (notably sucrose) from sources (photosynthetic organs) to sinks (growth points and storage tissues/organs) involves both active and passive processes. Active processes are primarily associated with the loading and unloading phases of phloem transport. At source tissues, sucrose is actively transported into companion cells or sieve elements against a concentration gradient via membrane-bound sucrose/proton symporters, driven by ATP-dependent proton pumps [83]. For example, in apoplastic phloem loading, the proton-motive force established by H^+^-ATPases energizes sucrose uptake into phloem loading cells, resulting in high phloem solute concentrations and concomitant osmotic water influx from accompanying xylem vessels, generating turgor pressure [84,85,86]. During unloading, sink tissue uptake may also involve energy-dependent transport across plasma membranes [87].

Passive processes operate during both loading and long-distance transport phases. In certain species, particularly woody plants, sucrose and other small solutes diffuse symplastically (in the continuous cytoplasm running between cells via plasmodesmata) into the sieve-element/companion cell complex, which does not require direct metabolic energy input at the loading interface [88]. Once a turgor pressure gradient between source (high turgor pressure) and sink (low turgor pressure) is established, bulk mass flow of phloem sap occurs along the sieve tubes by pressure-flow (known as the Münch mechanism) [86]. Additionally, unloading into sink tissues may occur via diffusion or bulk-flow through plasmodesmata (“convective unloading”) without exclusive reliance on energy-driven steps [89].

Thus, the active components of phloem functioning (loading/unloading via transporters and proton pumps) establish the solute and pressure gradients driving passive bulk flows. The passive components (symplastic diffusion, bulk flow under pressure gradients) carry assimilates between source and sink once those gradients exist. Together, these mechanisms enable the regulated and efficient transport of carbon and other solutes essential for plant metabolism and growth.

Neither xylem or phloem is entirely passive (nor entirely active), and vascular translocation in plants involves bulk flows described by classical physics, either initiated or regulated by active mechanisms involving the Brownian (thermodynamic) functioning of molecular machinery, particularly enzymatic proton pumps. Thus, thermodynamic effects at the level of cells and molecules scale up to maintain bulk flows that overcome diffusion limitations to size, resulting in some of the largest known organisms.

## 9. Translocation Within Large Heterokonts

Heterokonts (formally, Stramenopiles) are fundamentally different from plants because, aside from being defined by cells possessing two different types of flagella, photosynthetic taxa possess chloroplasts with four bounding membranes, indicating two endosymbiosis events during their evolutionary history (plant chloroplasts have two membranes and also differ in terms of photosynthetic pigments, lacking chlorophyll c and fucoxanthin). The plastid-bearing and thus photosynthetic stramenopiles (the ochrophytes) diverged between 1298 and 622 Ma [90]. Within the brown algae (Phaeophyceae), kelps (order Laminariales; ~31 Ma [91]) evolved complex systems to meet the demands of their extremely large size. They do not possess the true vascular tissues of tracheophytes but do exhibit convergent evolution of specialized vascular-like tissues that facilitate the transport of water, nutrients, and photosynthates. These tissues include trumpet-shaped sieve elements, analogous to the phloem in vascular plants, involved in the long-distance transport of organic compounds. Indeed, vascular optimization in kelp is considered convergent with tracheophytes [92], highlighting that although kelps are only distantly related, they have developed an equivalent transport system.

In the species *Nereocystis luetkeana* (K.Mertens) Postels & Ruprecht 1840, the movement of fluorescent dyes through sieve-tube networks demonstrates source-to-sink bulk flow and even pressure-driven reversal of flow across sieve plates, which is mechanistically equivalent to phloem transport in tracheophytes (sources are photosynthetically active blades; sinks are growth zones at stipe bases or holdfasts) [92,93]. These sieve elements are embedded in a gelatinous extracellular matrix composed largely of alginate polysaccharides, which isolates them from neighboring cells and thus provides a biomechanical buffer. Indeed, the internal tissues are organized such that the meristoderm (photosynthetic outer layer) and cortex overlie a central medulla where the sieve-element network resides. Transport velocities in kelp sieve elements have been reported in the range of tens to hundreds of mm h^−1^ (for example, a mean of 430 mm h^−1^ determined by following the motion of fluorescent dye markers [92]; this study also discussed estimates of 50–780 mm h^−1^ in the wider literature, determined using methods such as ^14^C-labeling of photoassimilates).

The sieve tubes form branched networks with junctions and lateral connections between parallel sieve tubes; some appear active while others are not, which is suggestive of flow regulation within the network. Unlike vascular plants, kelps lack water-conducting vessels equivalent to xylem and companion cells associated with sieve elements; the structural analogue is more basic.

Further research by Knoblauch et al. [93] revealed that the cell walls of sieve tubes in kelps can swell reversibly in response to changes in turgor pressure; this wall swelling may act as a turgor-generation and buffering mechanism, providing a regulatory dimension to flow control. While organic-carbon transport is one major function, kelps must also assimilate nutrients (nitrogen, phosphorus, and trace metals) from seawater and distribute these internally to growth zones and can take up multiple forms of nitrogen (nitrate, ammonium, urea) and vary their uptake depending on internal nitrogen status, thus linking external nutrient dynamics to internal transport needs.

Thus, analogous to plant phloem, the translocation system of kelp involves pressure flows operating over scales often measured in meters, but the physical forces generating pressure gradients are essentially due to energy-demanding cellular activity involving biomolecules that are ultimately driven by thermodynamics—again, macroscopic ‘classical’ physics driven by nanoscopic molecular processes.

## 10. Honorable Mentions

Aside from fungi, slime molds, animals, green plants, and heterokonts, there are a few other major eukaryotic lineages in which some large organisms and multicellular-like forms have evolved, although these are exceptions rather than the rule within their lineages.

Marine xenophyophores (SAR clade: Rhizaria: Foraminifera: Xenophyophorea) include one of the largest single-celled organisms (at least in terms of having a diameter measured in the tens of centimeters, even though the volume of the body or ‘test’ is mainly composed of sediment material): *Syringammina fragilissima* Brady 1883. This has a very particular biology, as a fragile benthic analog to a slime mold, and, functionally, can similarly be summarized as coenocytic [94].

Haplozoon protists such as *Haplozoon axiothellae* Siebert are dinoflagellate (SAR clade: Alveolata: Dinoflagellata: Haplozoonaceae) endoparasites of polychaetes. They give the appearance of being multicellular, to the extent that they were originally misclassified as a ‘simple’ ‘animal’ (literally ‘Haplozoon’). The apparent multicellularity actually represents a single multinucleate cell that develops to create internal septa (membranes) and specialized compartments, resembling a string of several cells with slightly different forms and functions. Interestingly, the protoplasm remains continuous throughout the compartments, and from a functional point of view this has been described as syncytial [95].

Crucially, in the cases mentioned here, internal long-distance or compartment-to-compartment transport is managed by the cytoskeleton and cytoplasmic streaming, and so uses the same physical processes as other eukaryotes and is not fundamentally different (although the lifestyles and evolutionary histories of these organisms certainly are). Again, diffusion limitations are overcome and larger size is facilitated because molecular action generates flow between compartments (albeit only at the level of cytoplasmic streaming; dedicated transport tissues are not involved in these cases).

## 11. Discussion

The view that emerges here is that living systems scale up by superimposing dedicated transport systems upon an underlying ‘Brownian’ system of molecular motion: from purely Brownian/diffusion to motor-driven transport on the cytoskeleton, cytoplasmic streaming, long-distance advection (sometimes including peristaltic mechanisms), up to organ- or organism-level bulk-flow systems. Vascular systems in multicellular organisms fundamentally use sub-cellular, thermodynamic (and ultimately quantum-mechanic) processes to either induce or regulate classical pressure gradients, with macroscopic classical processes ultimately emerging from quantum mechanical forces. Thus, a key evolutionary stage beyond simple multicellularity was the emergence of bulk-flow transport networks that extended the influence of Brownian and active transport processes into the macroscopic domain, overcoming diffusion limitations and thereby enabling large size and integrated physiology.

While network integration is a universal concept across scales, the specific challenge for size and physiology is to maintain transport rates and coupling across distances. The evolution of bulk-flow networks aggregated metabolic units (cells or tissues) into large coherent organisms by providing a high-capacity means of connecting modules. This makes bulk flow a key ‘next step’ following multicellularity.

However, several limitations and open questions remain. First, the detailed mechanisms by which bulk-flow networks evolve remain under-explored. For example, how do vascular systems or circulatory networks emerge from simpler transport systems? What regulatory and developmental changes are required? Second, while systems-biology frameworks have addressed network integration and dynamics, they rarely incorporate explicit transport flows (especially advection) in large-scale models of organismal physiology. There is a need for models that integrate transport networks with regulatory and metabolic networks in evolutionary contexts. Third, size is still constrained: even organisms with bulk flow face limits (e.g., diffusion from vessels into tissues, constraints of pump power, and network maintenance costs). Exploring the trade-offs between transport capacity, metabolic cost, and organism size remains fertile ground.

Future research directions include (a) comparative studies across lineages of the evolution of transport networks (e.g., plants vs. animals vs. colonial algae) to identify common design principles; (b) modeling frameworks in systems biology that explicitly integrate advection and network flows with regulation and metabolism; (c) synthetic biology or bio-engineering of model organisms or tissues to test how transport network topology and capacity influence size and integration; (d) palaeobiological and phylogenetic analyses that explore the timing and constraints of transport-network emergence in major evolutionary transitions.

## 12. Conclusions

This review collates evidence that bulk flow networks are a critical evolutionary innovation that enable eukaryotes to overcome size constraints. Despite disparate forms and specific mechanisms that generate pressure differentials within each clade (i.e., phylogenetic constraints), pressure gradients are consistently regulated by molecular-scale forces that exploit Brownian motion and thermodynamic phenomena. These phenomena are well documented separately at different scales (molecular, cellular, and intracellular) and for each group of organisms. However, the overview provided here suggests that macroscopic physiology (integration of metabolism with biological processes occurring throughout the organism) represents a layer of organization built upon this shared basic organismal functioning. This ‘Brownian baseline’ provides a common point of reference for considering the evolution of transport systems and large size in macroscopic organisms in general.

## Figures and Tables

**Figure 1 life-15-01914-f001:**
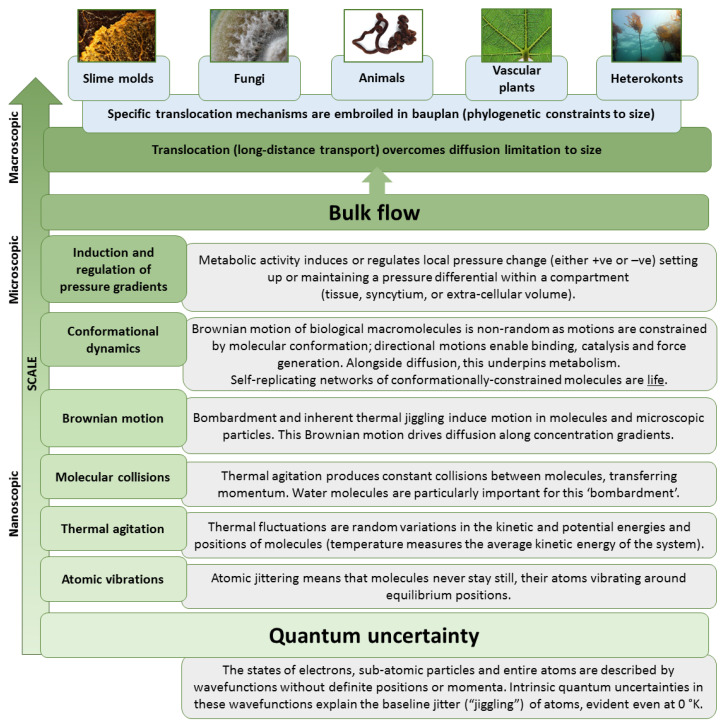
The relationship between quantum mechanics and bulk flow translocation systems in macroscopic eukaryotes. Images are not AI-generated and are published under a Creative Commons Attribution-Share Alike 3.0 or 4.0 license at commons.wikimedia.org (accessed on 4 November 2025) and compiled using PowerPoint (Microsoft Corp., Redmond, WA, USA). Individual image credits (from left to right): *Physarum polycephalum* (Helen Ginger); *Pyricularia oryzae* (Aleksandra50); *Lineus longissimus* (Bruno C. Vellutini); *Ficus carica* (Jon Richfield); *Nereocystis luetkeana* (Chris Teague).

**Figure 2 life-15-01914-f002:**
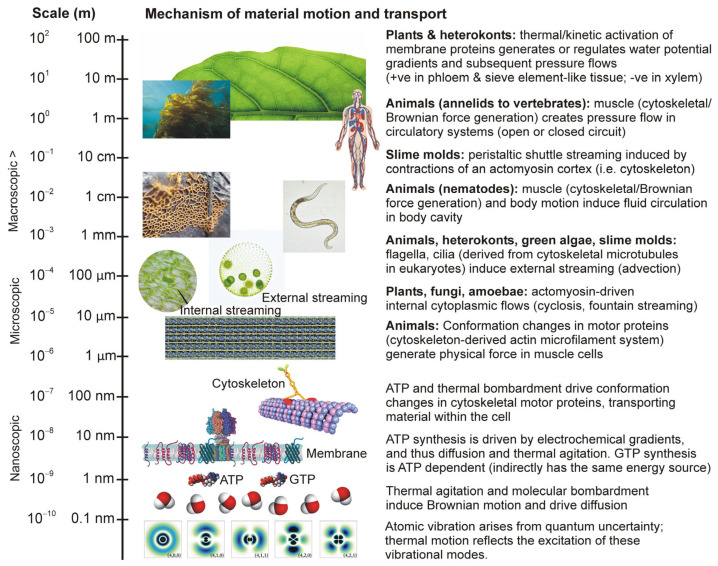
Physical scales of operation for the specific mechanisms of material motion and transport of large eukaryotes. Disparate adaptations, cell types, and tissue systems in different taxa share the same quantum mechanical underpinnings, with molecular motions and conformation state changes translating quantum motion into classical motion. Images are not AI-generated and are either public domain or published under a Creative Commons Attribution-Share Alike 3.0 or 4.0 license at commons.wikimedia.org (accessed on 4 November 2025) and compiled using CorelDraw 9 (Corel Corporation). Individual image credits (from top to bottom of page): backlit *Ficus elastica* leaf (Julian Herzog); *Nereocystis luetkeana* (Chris Teague); human organ systems (Loneshieling); slime mold (Nativeplants garden); Nematoda (Vyzhdova V); external streaming of *Volvox aureus* (Matthew Herron); internal streaming of *Elodea canadensis* cells (Daemon Canchig); molecular model of the sarcomere in the A band (Dirk Gently PI); kinesin on a cytoskeletal microtubule (Tomoki Fukushima); membrane proteins (Eunice Laurent); ATPsynthase (Bone117).

**Table 1 life-15-01914-t001:** The physical scale at which internal transport shifts from diffusion to pressure flow systems in Metazoa (note that this is not a simple, direct evolutionary transition via anagenesis; circulatory systems have evolved independently via convergent evolution in different metazoan taxa).

Physical Scale (mm)	Example Organisms	Body Cavity Type	Traits	Circulatory System
0.1–10	Flatworms (*Planaria*; *Platyhelminthes*)	Acoelomate	No internal cavity (solid tissue)	Diffusion only
0.5–500	Nematodes (*Nematoda*), Rotifers	Pseudo-coelomate	Cavity not fully lined with mesoderm	Circulation by body movement (pseudocoelomic fluid)
1>	Earthworms (*Annelida*), Mollusks, Arthropods, Vertebrates	Coelomate	Cavity fully lined with mesoderm	True circulatory system (open or closed)

## Data Availability

Not applicable.

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
