# Peer review of "The Evolution of Large Organism Size: Disparate Physiologies Share a Foundation at the Smallest Physical Scales"

_life, 2025, doi:10.3390/life15121914_

Round 1

Reviewer 1 Report

Comments and Suggestions for Authors

The manuscript presents an ambitious and conceptually interesting framework linking quantum-mechanical phenomena, thermal agitation, Brownian motion, and macroscopic biological processes, with illustrative examples ranging from enzyme catalysis to muscle contraction to whole-organism cytoplasmic streaming. The manuscript is clearly written, draws on a broad body of literature, and offers a perspective that can stimulate interdisciplinary discussion. Based on the scientific content and clarity of exposition, I believe the manuscript is suitable for publication pending one minor request.

One aspect that may be confusing for biological readers is the manuscript’s repeated use of the term “quantum” to describe the origins of macroscopic biological motion. While it is conceptually correct that thermal agitation and Brownian dynamics arise from underlying quantum-mechanical principles, several sentences blur the distinction between (i) the quantum origin of thermal fluctuations and (ii) direct quantum coherence or quantum biological effects at the organismal or tissue scale—which is not what the cited literature demonstrates. To aid clarity, the authors should consider including a simple descriptive figure or flowchart illustrating the conceptual hierarchy—quantum → thermal agitation → molecular fluctuations → mesoscale transport → macroscale motion—to visually unify the examples discussed and make the mechanistic progression clearer for a broad biological audience.

Author Response

[Comment 1]

The manuscript presents an ambitious and conceptually interesting framework linking quantum-mechanical phenomena, thermal agitation, Brownian motion, and macroscopic biological processes, with illustrative examples ranging from enzyme catalysis to muscle contraction to whole-organism cytoplasmic streaming. The manuscript is clearly written, draws on a broad body of literature, and offers a perspective that can stimulate interdisciplinary discussion. Based on the scientific content and clarity of exposition, I believe the manuscript is suitable for publication pending one minor request.

One aspect that may be confusing for biological readers is the manuscript’s repeated use of the term “quantum” to describe the origins of macroscopic biological motion. While it is conceptually correct that thermal agitation and Brownian dynamics arise from underlying quantum-mechanical principles, several sentences blur the distinction between (i) the quantum origin of thermal fluctuations and (ii) direct quantum coherence or quantum biological effects at the organismal or tissue scale—which is not what the cited literature demonstrates. To aid clarity, the authors should consider including a simple descriptive figure or flowchart illustrating the conceptual hierarchy—quantum → thermal agitation → molecular fluctuations → mesoscale transport → macroscale motion—to visually unify the examples discussed and make the mechanistic progression clearer for a broad biological audience.

  • [Response 1]: I genuinely appreciate this suggestion: I have now produced a new figure which shows (as a flow-chart) the various passages from quantum to classical motions, and I think it definitely helps to summarize the main concepts I am attempting to communicate. The subject matter does span a wide range of scales and areas of study, so a visual summary is indeed a big help. Thank you.

Reviewer 2 Report

Comments and Suggestions for Authors

This manuscript explores how the evolution of large organismal size across different taxa may be underpinned by processes operating at the smallest physical scales. The author proposes that all macroscopic transport systems (circulatory, vascular, cytoplasmic streaming, etc.) ultimately rely on molecular-scale (quantum-driven) dynamics, effectively linking quantum-level phenomena (thermal motion, Brownian dynamics) to classical bulk-flow physiology. The review surveys a broad range of organisms, from slime molds and fungi to tracheophyte plants and animals, to argue that the emergence of bulk flow networks was a critical evolutionary step enabling larger body size and integrated multicellular physiology. Overall evaluation: The topic is ambitious and cross-disciplinary, aiming to unite principles of physics with evolutionary biology. The manuscript is rich in ideas and attempts a novel synthesis. However, as an evolutionary biologist, I find that several aspects require clarification and refinement. In its current form, the manuscript’s conceptual framework and terminology may be confusing to readers, and parts of the narrative risk appearing teleological or speculative. I commend the author’s interdisciplinary approach, but significant revisions are needed to improve the clarity, precision, and evolutionary grounding of the arguments.

Major comments:

  1. Conceptual foundation: integration of quantum processes into evolutionary transitions: The central premise is that quantum-scale processes laid the foundation for evolutionary transitions to larger size. While it is true that molecular (thermodynamic) processes underpin all biology, the manuscript needs to more clearly articulate why framing this as a “quantum-to-classical” evolutionary transition provides new insight. Currently, the argument can appear circular: essentially all biology emerges from physics at small scales, so of course large organisms ultimately rely on quantum phenomena. To avoid this seeming trivial, clarify what is novel or insightful about viewing the evolution of bulk flow through a quantum lens. For example, emphasize how specific evolutionary innovations harnessed molecular-scale randomness to drive organized transport (e.g. the evolution of contractile molecular motors in cardiac muscle translating Brownian-scale events into organ-scale blood flow). If “quantum processes” here mostly refer to thermal molecular motion and stochastic enzyme activity, consider using more accessible terms like “molecular-scale” or “stochastic thermodynamic” processes in the biological context. This would ground the concept in well-understood biochemical principles while avoiding overreach. As it stands, the manuscript sometimes gives the impression of stretching the physics terminology for effect, rather than strengthening the evolutionary argument. The idea that organized bulk flow was a key evolutionary step beyond simple multicellularity is compelling – indeed, it aligns with the notion of a major evolutionary transition enabling new levels of complexity – but it should be presented in a way that biologists can readily grasp. I encourage the authors to better justify how the quantum perspective advances our understanding of size/physiology evolution, or else dial back the emphasis and focus on molecular mechanisms (e.g. motor proteins, diffusion limits) in more conventional terms.
  2. Clarity of terminology (e.g. “quantum-to-classical transition”, “Brownian foundation of physiology”): Several phrases and terms used in the manuscript are potentially confusing or non-standard and should be defined or rephrased for clarity. The phrase “quantum-to-classical transition” is central to the paper, yet readers might not intuitively understand it in a biological context. In physics, this phrase refers to the emergence of classical behavior from quantum rules (often via decoherence). The manuscript appears to use it as a metaphor for the scale shift from molecular processes to bulk phenomena, for example, the text describes long-distance pressure-driven flows emerging from molecular agitation, “essentially a transition from quantum to classical mechanics”. I recommend explicitly defining this term early on (perhaps in the Introduction or a dedicated subsection) in plain language. For instance: “we refer to a ‘quantum-to-classical transition’ in physiology to mean the scale at which molecular (thermodynamic, stochastic) processes give rise to emergent bulk properties describable by classical physics (fluid dynamics)”. This will ensure readers do not misinterpret it as something like quantum evolution or quantum biology in the sense of coherence phenomena (which is not the focus here). Similarly, phrases like “Brownian foundation of physiology” and “quantum origins of bulk flow” should be clarified. It may be better to say “molecular-scale (Brownian) motion as the foundation” or “molecular origins of bulk flow” to avoid conflating quantum mechanics per se with biology. The manuscript does a thorough job describing how molecular agitation and motor proteins drive bulk processes – simply make sure the terminology used to label these concepts is accessible and precise. If these terms are your own coinage, consider placing them in quotes and explaining them, or substituting more standard descriptions. Clear, jargon-free terminology will greatly improve the readability of the paper.
  3. Empirical synthesis across taxa: the review impressively spans a wide range of life forms (animals, plants, fungi, algae, slime molds) to support the thesis that large organisms share similar transport solutions. This broad comparative approach is a strength. To enhance it further, ensure that each section not only describes the taxon-specific details but explicitly ties back to the unifying concept. For example, when discussing plant vascular systems versus animal circulatory systems, highlight the common principle (bulk flow overcoming diffusion limits) as well as any instructive differences. The manuscript sometimes reads as a catalog of examples; it would benefit from a bit more synthetic commentary that compares and contrasts these systems. For instance, are there size or distance thresholds common to all lineages where diffusion alone fails (the ~100 µm scale is mentioned for metazoans)? If so, emphasize that convergence. Are there cases where large size was achieved without evolving extensive transport networks (e.g. large coenocytic algae or colonial organisms)? A brief acknowledgment of such exceptions or alternate strategies would strengthen the evolutionary argument. Currently, the examples are well-chosen, and the inclusion of less commonly cited cases (e.g. myxogastrid plasmodial streaming, fungal hyphal transport, giant algal siphons) is commendable. Just be sure to consistently relate these back to the central thesis in the text. This will help the reader appreciate the integration across taxa, rather than seeing each section in isolation. Adding one or two summary sentences at the end of major sections (or in the Discussion) synthesizing how disparate taxa illustrate the same foundational principles would reinforce the manuscript’s message.
  4. Teleological and anthropomorphic language: this is an important one. I encourage the authors to review the wording throughout to avoid any unintended teleology or anthropomorphism. Phrases that imply purpose or intent in evolution should be rephrased in terms of natural selection or functional consequence. For example, the manuscript states that a primitive circulatory system appeared “as a strategy to overcome the diffusion limit over larger transport distances”. This wording can be read as if evolution consciously devised a strategy. It would be better to say something along the lines of: “likely evolved in response to the diffusion limit imposed by larger body size”. This maintains a scientific, non-teleological tone. Similarly, be cautious with words like “in order to”, “to enable”, or “to allow” when describing evolutionary developments, these can often be rephrased as “which enabled X” or “resulting in Y, which conferred advantage Z”. Another example is describing plants as “investing” energy in creating osmotic gradients – while metaphorically clear, it borders on anthropomorphic phrasing. Consider alternatives like “plants expend metabolic energy to generate osmotic gradients…”. The overall narrative should be about processes shaped by selection rather than goal-directed strategies. These are mostly matters of tone, but they are important in evolutionary writing. Ensuring strict Darwinian phrasing will make the paper more rigorous. A careful copy-edit for such instances of teleological language is recommended.
  5. The manuscript presents a grand narrative of how transport systems enabled large organismal size. While this broad story is engaging, at times the explanation of mechanisms and causal specifics is lacking or uneven. For a review in Life, readers will expect both the “big picture” and some discussion of how these transitions occurred. Currently, some key transitions (e.g. the origin of animal circulatory systems, or the evolution of plant vascular tissue) are described in a historical sense but not deeply in a mechanistic sense. For instance, what developmental or genetic innovations might have been involved in the first primitive circulatory vessels? Are there known intermediate forms or steps (even hypothetical) between diffusion-only organisms and those with rudimentary bulk flow? The text notes that detailed mechanisms for evolving bulk flow are under-explored (and rightly highlights this as an open question). To strengthen the paper, I suggest incorporating a bit more discussion on possible mechanisms or citing studies that have addressed evolutionary precursors of transport networks. Even speculative discussion (clearly labeled as such) could be valuable, for example, mention gene regulatory networks involved in vasculature development or the co-option of contractile proteins for pumping. In the plant section, one could briefly mention how early land plants (e.g. Cooksonia) show simple vascular strands, hinting at how selection for height may have driven conductive tissue evolution. In animals, perhaps mention the multiple independent origins of circulatory systems (as you do) and relate this to modular evolution of hearts or vessels. The goal is to provide some mechanistic plausibility to complement the narrative. Currently, the narrative is interesting but can feel abstract; adding mechanistic context (even if drawn from modern analogues or theory) will ground the story in concrete evolutionary biology. This also helps address the “how” questions: not just that bulk flow is beneficial, but how lineages actually achieved it stepwise.
  6. The manuscript would benefit from a thorough edit for clarity and conciseness. Some sentences are quite long or densely packed with concepts. Try to break complex sentences into smaller ones, and explicitly define terms when first used. For example, the opening lines of the abstract/intro are philosophically interesting but hard to parse on first read – simplifying the language there would ensure you don’t lose readers early. Likewise, when introducing interdisciplinary concepts (quantum physics, thermodynamics), do so gradually and in relation to biology. Ensure consistent use of terms: at times “quantum mechanical” is repeated very frequently, consider if in some places simply “molecular” or “nanoscale” would convey the meaning with less potential confusion. In terms of precision: avoid over-generalizations like “all macroscopic physiologies are underpinned by quantum mechanics” without qualification. While essentially true in a physical sense, biologically it might read as hyperbole. Instead, specify that molecular processes (rooted in physics) underpin physiology, a subtle but important framing to keep the biology in focus. The manuscript is well-referenced, which is excellent; just make sure each claim is accurately supported by the citation given, and add clarifying context where needed so that a reader understands the evidence being cited. I noticed the reference list includes a mix of classical physics papers and modern biological studies, ensure that in the text, these are integrated smoothly. For instance, when citing decoherence theory or de Broglie, relate it immediately to your point about biology, so a non-physicist reader isn’t left wondering why that reference is there. A careful line-by-line edit focusing on readability and exact wording will greatly improve the manuscript’s impact. I also recommend having a colleague or two from different backgrounds read a draft to flag any confusing passages, given the inherently cross-disciplinary nature of the paper.

In the Discussion, the manuscript identifies open questions and research directions, which is excellent. I particularly liked the forward-looking points about modeling and comparative studies. One suggestion is to ensure the Conclusions section clearly encapsulates your main thesis in a concise way. A reader should come away with the take-home message that “bulk flow networks were a critical evolutionary innovation enabling life to overcome size constraints, and despite disparate forms (vessels, xylem, streaming), these all rely on the same fundamental molecular-scale forces.” Currently the Conclusion does emphasize this idea. Just make sure it’s stated in the simplest, most accessible terms to leave a strong impression. Lastly, double-check the consistency of units and taxonomy names (all in italics where needed, etc.), and fix any minor typos. The quality of writing is generally high, but polishing these details will help readers focus on the fascinating ideas you are presenting.

This manuscript tackles a fascinating intersection of physics and evolutionary biology and provides a broad comparative treatment of how large organisms manage internal transport. The revisions I suggest are aimed at improving clarity and rigor: refining terminology, strengthening evolutionary explanations, and avoiding language pitfalls. I believe that with these changes, the paper will be much clearer and more convincing to the Life readership. I encourage the author to carefully address each of the above points. The effort will be worthwhile, as this synthesis has the potential to stimulate cross-disciplinary thinking about the evolution of physiological complexity.

Author Response

[Comment 1]

This manuscript explores how the evolution of large organismal size across different taxa may be underpinned by processes operating at the smallest physical scales. The author proposes that all macroscopic transport systems (circulatory, vascular, cytoplasmic streaming, etc.) ultimately rely on molecular-scale (quantum-driven) dynamics, effectively linking quantum-level phenomena (thermal motion, Brownian dynamics) to classical bulk-flow physiology. The review surveys a broad range of organisms, from slime molds and fungi to tracheophyte plants and animals, to argue that the emergence of bulk flow networks was a critical evolutionary step enabling larger body size and integrated multicellular physiology. Overall evaluation: The topic is ambitious and cross-disciplinary, aiming to unite principles of physics with evolutionary biology. The manuscript is rich in ideas and attempts a novel synthesis. However, as an evolutionary biologist, I find that several aspects require clarification and refinement. In its current form, the manuscript’s conceptual framework and terminology may be confusing to readers, and parts of the narrative risk appearing teleological or speculative. I commend the author’s interdisciplinary approach, but significant revisions are needed to improve the clarity, precision, and evolutionary grounding of the arguments.

Major comments:

  1. Conceptual foundation: integration of quantum processes into evolutionary transitions: The central premise is that quantum-scale processes laid the foundation for evolutionary transitions to larger size. While it is true that molecular (thermodynamic) processes underpin all biology, the manuscript needs to more clearly articulate why framing this as a “quantum-to-classical” evolutionary transition provides new insight. Currently, the argument can appear circular: essentially all biology emerges from physics at small scales, so of course large organisms ultimately rely on quantum phenomena. To avoid this seeming trivial, clarify what is novel or insightful about viewing the evolution of bulk flow through a quantum lens. For example, emphasize how specific evolutionary innovations harnessed molecular-scale randomness to drive organized transport (e.g. the evolution of contractile molecular motors in cardiac muscle translating Brownian-scale events into organ-scale blood flow). If “quantum processes” here mostly refer to thermal molecular motion and stochastic enzyme activity, consider using more accessible terms like “molecular-scale” or “stochastic thermodynamic” processes in the biological context. This would ground the concept in well-understood biochemical principles while avoiding overreach. As it stands, the manuscript sometimes gives the impression of stretching the physics terminology for effect, rather than strengthening the evolutionary argument. The idea that organized bulk flow was a key evolutionary step beyond simple multicellularity is compelling – indeed, it aligns with the notion of a major evolutionary transition enabling new levels of complexity – but it should be presented in a way that biologists can readily grasp. I encourage the authors to better justify how the quantum perspective advances our understanding of size/physiology evolution, or else dial back the emphasis and focus on molecular mechanisms (e.g. motor proteins, diffusion limits) in more conventional terms.

  • [RESPONSE 1]: This comment, concerning why the current review across scales is novel, got me thinking about how I was communicating the main message, particularly at the end of the Abstract. It didn’t really showcase the phylogenetic effect (i.e. that the different body forms are intimately linked to how transport is scaled up from the shared underlying cellular mechanisms). I have generally changed the language to a more biological wording, as suggested, but mainly I substituted the last part of the abstract with the following, which I think also ties in better with the title:

“Macroscopic physiologies are underpinned by Brownian molecular thermodynamics and thus quantum mechanics; the apparently disparate physiologies of large organisms share a fundamental operating principle at small scales. However, the specific translocation mechanisms that extend this functioning to larger scales are embroiled in bauplans, respresenting phylogenetic constraints to body size. ”

From this, and similar changes in wording in the main text, I hope that the novelty of this perspective is evident (i.e. the transition from a shared ‘Brownian’ transport system to macroscopic transport is a major transition, constrained by bauplan).

[Comments 2]

  1. Clarity of terminology (e.g. “quantum-to-classical transition”, “Brownian foundation of physiology”): Several phrases and terms used in the manuscript are potentially confusing or non-standard and should be defined or rephrased for clarity. The phrase “quantum-to-classical transition” is central to the paper, yet readers might not intuitively understand it in a biological context. In physics, this phrase refers to the emergence of classical behavior from quantum rules (often via decoherence). The manuscript appears to use it as a metaphor for the scale shift from molecular processes to bulk phenomena, for example, the text describes long-distance pressure-driven flows emerging from molecular agitation, “essentially a transition from quantum to classical mechanics”. I recommend explicitly defining this term early on (perhaps in the Introduction or a dedicated subsection) in plain language. For instance: “we refer to a ‘quantum-to-classical transition’ in physiology to mean the scale at which molecular (thermodynamic, stochastic) processes give rise to emergent bulk properties describable by classical physics (fluid dynamics)”. This will ensure readers do not misinterpret it as something like quantum evolution or quantum biology in the sense of coherence phenomena (which is not the focus here). Similarly, phrases like “Brownian foundation of physiology” and “quantum origins of bulk flow” should be clarified.

  • [Response 2]: Although I cannot find these specific phrases in the text, I understand the point: to use phrasing that places more emphasis on biological molecules and thermodynamics and less on quantum mechanics, in order to be clear to biologists. So although I still explain thermodynamics as emerging from quantum mechanics, I have now moderated any references to this and favoured thermodynamic and Brownian terminology. Section 2 is explicitly an explanation of quantum effects, so it is inevitable that I talk specifically about this here, but I have moderated language from that point onwards, for example Section 3 now starts with “stochastic thermodynamic”.

It may be better to say “molecular-scale (Brownian) motion as the foundation” or “molecular origins of bulk flow” to avoid conflating quantum mechanics per se with biology. The manuscript does a thorough job describing how molecular agitation and motor proteins drive bulk processes – simply make sure the terminology used to label these concepts is accessible and precise. If these terms are your own coinage, consider placing them in quotes and explaining them, or substituting more standard descriptions. Clear, jargon-free terminology will greatly improve the readability of the paper.

àResponse: I have revised the text for clarity.

[Comment 3]

  1. Empirical synthesis across taxa: the review impressively spans a wide range of life forms (animals, plants, fungi, algae, slime molds) to support the thesis that large organisms share similar transport solutions. This broad comparative approach is a strength. To enhance it further, ensure that each section not only describes the taxon-specific details but explicitly ties back to the unifying concept. For example, when discussing plant vascular systems versus animal circulatory systems, highlight the common principle (bulk flow overcoming diffusion limits) as well as any instructive differences. The manuscript sometimes reads as a catalog of examples; it would benefit from a bit more synthetic commentary that compares and contrasts these systems. For instance, are there size or distance thresholds common to all lineages where diffusion alone fails (the ~100 µm scale is mentioned for metazoans)?

  • [Response 3]: With regard to the value of 100 µm, it was originally shown in section 4 (“Diffusion limitation to organism size”) and comes from the citation [49], which was a general citation. Indeed, the paper showed examples from various eukaryotes, not just metazoans. Later, in the section on metazoans (line 322) the size threshold of 100 µm was mentioned again in this context, but without the citation. Evidently by repeating the value in the metazoa section, in the mind of the reader it becomes associated with metazoa. My job as the writer is to avoid confusion and communicate effectively, so I have decided to make two revisions here. First, I have explicitly qualified the value by stating that it is “a value determined by diffusion that holds true for all major eukaryotic clades”. Second, while the paper I originally cited was a general discussion of cell size, I have replaced this with a perhaps more rigorous study that specifically compares cells of plants, animals and fungi, including a discussion of the limits to cell size. This is also more recent (2023). Additionally, I think that the new flow chart (Fig. 1) suggested by Reviewer 1 also highlights the common principle shared by different eukaryote clades, helping address the concern raised here.

If so, emphasize that convergence. Are there cases where large size was achieved without evolving extensive transport networks (e.g. large coenocytic algae or colonial organisms)? A brief acknowledgment of such exceptions or alternate strategies would strengthen the evolutionary argument.

  • [Response]: I had included a paragraph about the genus Caulerpa as an example of large coenocytic algae lacking a transport network.

Currently, the examples are well-chosen, and the inclusion of less commonly cited cases (e.g. myxogastrid plasmodial streaming, fungal hyphal transport, giant algal siphons) is commendable. Just be sure to consistently relate these back to the central thesis in the text. This will help the reader appreciate the integration across taxa, rather than seeing each section in isolation. Adding one or two summary sentences at the end of major sections (or in the Discussion) synthesizing how disparate taxa illustrate the same foundational principles would reinforce the manuscript’s message.

  • [Response]: Each section for each taxonomic group now has a summary ending paragraph that explicitly states that thermodynamic effects at the level of cells and molecules scale up to maintain bulk flows that overcome diffusion limitations to size.

[Comment 4]

  1. Teleological and anthropomorphic language: this is an important one. I encourage the authors to review the wording throughout to avoid any unintended teleology or anthropomorphism. Phrases that imply purpose or intent in evolution should be rephrased in terms of natural selection or functional consequence. For example, the manuscript states that a primitive circulatory system appeared “as a strategy to overcome the diffusion limit over larger transport distances”. This wording can be read as if evolution consciously devised a strategy. It would be better to say something along the lines of: “likely evolved in response to the diffusion limit imposed by larger body size”. This maintains a scientific, non-teleological tone. Similarly, be cautious with words like “in order to”, “to enable”, or “to allow” when describing evolutionary developments, these can often be rephrased as “which enabled X” or “resulting in Y, which conferred advantage Z”. Another example is describing plants as “investing” energy in creating osmotic gradients – while metaphorically clear, it borders on anthropomorphic phrasing. Consider alternatives like “plants expend metabolic energy to generate osmotic gradients…”. The overall narrative should be about processes shaped by selection rather than goal-directed strategies. These are mostly matters of tone, but they are important in evolutionary writing. Ensuring strict Darwinian phrasing will make the paper more rigorous. A careful copy-edit for such instances of teleological language is recommended.

  • [Response 4]: I have gone through the manuscript line by line and changed anything that could be interpreted as teleological. My background is in plant biology and ecology, where it is common to use terms such as ‘resource investment strategy’, but I can see that in an evolutionary context this is open not just to misinterpretation but also wilful misrepresentation, so it is doubly important to avoid teleology.

[Comment 5]

  1. The manuscript presents a grand narrative of how transport systems enabled large organismal size. While this broad story is engaging, at times the explanation of mechanisms and causal specifics is lacking or uneven. For a review in Life, readers will expect both the “big picture” and some discussion of how these transitions occurred. Currently, some key transitions (e.g. the origin of animal circulatory systems, or the evolution of plant vascular tissue) are described in a historical sense but not deeply in a mechanistic sense. For instance, what developmental or genetic innovations might have been involved in the first primitive circulatory vessels? Are there known intermediate forms or steps (even hypothetical) between diffusion-only organisms and those with rudimentary bulk flow? The text notes that detailed mechanisms for evolving bulk flow are under-explored (and rightly highlights this as an open question). To strengthen the paper, I suggest incorporating a bit more discussion on possible mechanisms or citing studies that have addressed evolutionary precursors of transport networks. Even speculative discussion (clearly labeled as such) could be valuable, for example, mention gene regulatory networks involved in vasculature development or the co-option of contractile proteins for pumping. In the plant section, one could briefly mention how early land plants (e.g. Cooksonia) show simple vascular strands, hinting at how selection for height may have driven conductive tissue evolution.

  • [Response 5]: I had included the example of polysporagiophytes, which is the group that includes Cooksonia pertoni, and presented the relationship between the simple vascular system and height. I have now explicitly added the name Cooksonia pertoni to this example.

[Comment]

In animals, perhaps mention the multiple independent origins of circulatory systems (as you do) and relate this to modular evolution of hearts or vessels. The goal is to provide some mechanistic plausibility to complement the narrative. Currently, the narrative is interesting but can feel abstract; adding mechanistic context (even if drawn from modern analogues or theory) will ground the story in concrete evolutionary biology. This also helps address the “how” questions: not just that bulk flow is beneficial, but how lineages actually achieved it stepwise.

  • [Response]: With regard to the evolution of cardiovascular systems, for example the acquisition of heart chambers, I have now added a paragraph supported by the citation of a review [77]. Effectively, a huge amount of detail is available from genetic/developmental studies of extant animal circulatory systems, with a central theme of conserved genes that are expressed differently or in different combinations to regulate different aspects of development. The cited review paints a very complex picture, but this is useful because the details of the evolution of circulatory systems in different groups are nonetheless supported by a simple operating principle of muscular contraction that pressurizes a compartment. This highlights the advantage of the current manuscript in pointing out that there is a shared basic operating principle, linked to smaller-scale physical phenomena, but also that adaptations scaling this up involve phylogenetic differences between different animal clades. I feel that the manuscript is much improved by informing the reader that these developmental details are known (even though it is well beyond the scope of this article to descend into the morass of pathways that are effectively reviewed elsewhere).

[Comment 6]

  1. The manuscript would benefit from a thorough edit for clarity and conciseness. Some sentences are quite long or densely packed with concepts. Try to break complex sentences into smaller ones, and explicitly define terms when first used. For example, the opening lines of the abstract/intro are philosophically interesting but hard to parse on first read – simplifying the language there would ensure you don’t lose readers early.

  • [Response 6]: I have revised the language to break up or simplify complex statements. This includes the first sentence of the Abstract and of the Introduction (with the caveat that as this defines living systems it can only be reduced to a certain point without losing meaning or precision).

[Comment]

Likewise, when introducing interdisciplinary concepts (quantum physics, thermodynamics), do so gradually and in relation to biology. Ensure consistent use of terms: at times “quantum mechanical” is repeated very frequently, consider if in some places simply “molecular” or “nanoscale” would convey the meaning with less potential confusion. In terms of precision: avoid over-generalizations like “all macroscopic physiologies are underpinned by quantum mechanics” without qualification. While essentially true in a physical sense, biologically it might read as hyperbole. Instead, specify that molecular processes (rooted in physics) underpin physiology, a subtle but important framing to keep the biology in focus. The manuscript is well-referenced, which is excellent; just make sure each claim is accurately supported by the citation given, and add clarifying context where needed so that a reader understands the evidence being cited. I noticed the reference list includes a mix of classical physics papers and modern biological studies, ensure that in the text, these are integrated smoothly. For instance, when citing decoherence theory or de Broglie, relate it immediately to your point about biology, so a non-physicist reader isn’t left wondering why that reference is there. A careful line-by-line edit focusing on readability and exact wording will greatly improve the manuscript’s impact. I also recommend having a colleague or two from different backgrounds read a draft to flag any confusing passages, given the inherently cross-disciplinary nature of the paper.

  • [Response]: I have generally edited for clarity, toned-down phrasing that could be interpreted as hyperbole and, as stated above, placed emphasis on thermodynamic, molecular and biological terminology rather than quantum mechanics.

[Comment]

In the Discussion, the manuscript identifies open questions and research directions, which is excellent. I particularly liked the forward-looking points about modeling and comparative studies. One suggestion is to ensure the Conclusions section clearly encapsulates your main thesis in a concise way. A reader should come away with the take-home message that “bulk flow networks were a critical evolutionary innovation enabling life to overcome size constraints, and despite disparate forms (vessels, xylem, streaming), these all rely on the same fundamental molecular-scale forces.” Currently the Conclusion does emphasize this idea. Just make sure it’s stated in the simplest, most accessible terms to leave a strong impression.

  • [Response]: I have modified the conclusions, and I think that the new Fig. 1 also helps summarise and communicate this message.

[Comment]

Lastly, double-check the consistency of units and taxonomy names (all in italics where needed, etc.), and fix any minor typos. The quality of writing is generally high, but polishing these details will help readers focus on the fascinating ideas you are presenting.

This manuscript tackles a fascinating intersection of physics and evolutionary biology and provides a broad comparative treatment of how large organisms manage internal transport. The revisions I suggest are aimed at improving clarity and rigor: refining terminology, strengthening evolutionary explanations, and avoiding language pitfalls. I believe that with these changes, the paper will be much clearer and more convincing to the Life readership. I encourage the author to carefully address each of the above points. The effort will be worthwhile, as this synthesis has the potential to stimulate cross-disciplinary thinking about the evolution of physiological complexity.

  • [Response]: I sincerely thank the reviewer – the suggestions were all highly pertinent, thoughtful and have helped improve the manuscript.

Round 2

Reviewer 2 Report

Comments and Suggestions for Authors

The revised manuscript shows that the authors have carefully considered and incorporated all substantive comments. The clarified structure, strengthened conceptual transitions, and updated discussion of recent advances have considerably improved the scholarly quality of the work. The revisions address the points previously raised and enhance both the precision and coherence of the review.